# Molecular Approaches and Echocardiographic Deformation Imaging in Detecting Myocardial Fibrosis

**DOI:** 10.3390/ijms231810944

**Published:** 2022-09-19

**Authors:** Andrea Sonaglioni, Gian Luigi Nicolosi, Elisabetta Rigamonti, Michele Lombardo, Lucia La Sala

**Affiliations:** 1IRCCS MultiMedica, 20138 Milan, Italy; 2Division of Cardiology, Policlinico San Giorgio, 33170 Pordenone, Italy

**Keywords:** TGF-beta signalling, myocardial fibrosis, microRNAs, speckle tracking echocardiography, myocardial strain, subclinical myocardial dysfunction, modified Haller index

## Abstract

The pathological remodeling of myocardial tissue is the main cause of heart diseases. Several processes are involved in the onset of heart failure, and the comprehension of the mechanisms underlying the pathological phenotype deserves special attention to find novel procedures to identify the site of injury and develop novel strategies, as well as molecular druggable pathways, to counteract the high degree of morbidity associated with it. Myocardial fibrosis (MF) is recognized as a critical trigger for disruption of heart functionality due to the excessive accumulation of extracellular matrix proteins, in response to an injury. Its diagnosis remains focalized on invasive techniques, such as endomyocardial biopsy (EMB), or may be noninvasively detected by cardiac magnetic resonance imaging (CMRI). The detection of MF by non-canonical markers remains a challenge in clinical practice. During the last two decades, two-dimensional (2D) speckle tracking echocardiography (STE) has emerged as a new non-invasive imaging modality, able to detect myocardial tissue abnormalities without specifying the causes of the underlying histopathological changes. In this review, we highlighted the clinical utility of 2D-STE deformation imaging for tissue characterization, and its main technical limitations and criticisms. Moreover, we focalized on the importance of coupling 2D-STE examination with the molecular approaches in the clinical decision-making processes, in particular when the 2D-STE does not reflect myocardial dysfunction directly. We also attempted to examine the roles of epigenetic markers of MF and hypothesized microRNA-based mechanisms aiming to understand how they match with the clinical utility of echocardiographic deformation imaging for tissue characterization and MF assessment.

## 1. Introduction

Myocardial fibrosis (MF) is a pathological remodelling process defined by the excessive accumulation in the myocardium of extracellular matrix (ECM) components, produced by cardiac fibroblasts, particularly collagen type 1, in response to an injury [1]. Even if MF is initially an adaptive mechanism, the excessive and continuous deposition of ECM reduces myocardial compliance [2], as well as affects the electrical properties of the myocytes [3]. MF, a well-recognized cause of morbidity and mortality [2], is the common final pathway of several ischemic and non-ischemic conditions promoting cardiac fibrosis, such as hypertensive, diabetic and idiopathic cardiomyopathy [1,4,5]. From a pathophysiological point of view, a dysregulation of ECM homeostasis leads to three main types of MF: (1) reactive interstitial fibrosis, (2) replacement fibrosis and (3) infiltrative interstitial fibrosis.

Reactive interstitial fibrosis (Rea-F) is an adaptive aspecific response characterized by an increased ECM deposition without alteration in a cardiomyocytes number [6], with the prolonged activation of pro-fibrotic growth factors, such as transforming growth factor-beta (TGF-β), connective tissue growth factor (CTGF) and fibroblast growth factor-2 (FGF2); this form of interstitial fibrosis is secondary to prolonged pressure overload, as in aortic stenosis and chronic hypertension [7] and/or prolonged volume overload, as in aortic regurgitation [8], and can be detected in a number of cardiomyopathies, including hypertrophic cardiomyopathy [9], dilated cardiomyopathy [10] and diabetic cardiomyopathy [11], as well as in heart failure with preserved ejection fraction [12].

In Reparative fibrosis (Rep-F), the death of cardiomyocytes and the rearrangement of collagen fibers are the key elements in stimulating fibrosis; these changes allow for the development of an organized fibrous scar tissue produced by myofibroblasts after cardiac injury, such as after myocardial infarction [4,13,14].

Moreover, another form of interstitial fibrosis, called Infiltrative (Inf-F), may be found during the progressive deposition of non-degradable matrix, such as in amyloidosis [15] and Fabry disease [16].

In this work, we sought to provide an innovative way to approach the atavistic problem for the assessment of MF, by describing the most invasive and non-invasive diagnostic tools. Recently, both biochemical and molecular markers were measured for MF diagnosis, but clear evidence for the echocardiographic imaging procedures has not already been demonstrated.

The aim of this review was to examine the main findings of the most important studies on MF, but few demonstrated a direct correlation between echocardiographic deformation imaging and molecular approaches. Therefore, we summarized the main invasive and non-invasive modalities for MF assessment, highlighting the usefulness of monitoring both molecular and imaging procedures in clinical practice.

## 2. Diagnosis of MF

### 2.1. The Gold Standard for Diagnosis of MF

The current “gold standard” for quantification of collagen deposition and diagnosing diffuse MF is the endomyocardial biopsy (EMB) [1,17,18]. Overall, EMB allows for direct microscopic assessment of the myocardial components and fibrotic changes and is particularly useful when other diagnostic tools are not successful and when a definitive diagnosis is needed to guide treatment, such as in the case of myocarditis, amyloidosis and sarcoidosis [19]. Its main limitations are related to some potential risks associated with its invasive nature (such as tricuspid valve injury, transient complete heart block, right bundle branch block and pericardial effusion) [20] and to the propensity for sampling deficit and errors, especially in cases of localized fibrosis [19]. Figure 1A.

Beyond the EMB, MF may be identified by a number of non-invasive techniques [21]. The most common non-invasive method for measuring MF is Cardiac Magnetic Resonance Imaging (CMRI), which employs gadolinium, an extracellular agent accumulating in interstitial fibrosis, oedema or infiltration areas [22], and chelating agents for myocardial T1 mapping, quantifying cardiac scar fibrosis [19,23] and extracellular volume fraction [24]. Despite the advantageous effects, CMRI is not free from adverse events such as intolerance phenomena to gadolinium-based contrast agent, especially increased risk of nephrogenic systemic fibrosis [25]; moreover, it is not recommended for patients with metallic implants and intracardiac devices [26], or for dyspnoic and claustrophobic patients [19]; in addition, CMRI is an expensive method which requires a long learning curve for both acquisition and analysis of the acquired images [14]. See Figure 1B.

### 2.2. Other Techniques for Diagnosing MF

Unlike EMB and cardiac MRI, serum biomarkers of MF (see Appendix A) seem to be more advantageous for diagnosis, therapeutic monitoring and prognosis. However, they should be considered mostly experimental, with variable sensitivity and specificity in different settings, and absolutely not widely implemented in clinical practice. A number of molecules, detectable in either the serum or plasma in humans, by using immunoassay methods, have recently been proposed as biomarkers of MF. Nevertheless, their relative experimental interpretation is difficult because some of them are inconclusive for MF. Indeed, the cardiac cell population sources of serum biomarkers are various and can include fibroblasts, endothelial cells, pericytes, and immune cells, each of which are characterized by specific pathways that they activate. Among the circulating biomarkers of MF, C-terminal telopeptide of collagen 1 (CITP), matrix metalloproteinases (MMPs) and their inhibitors (TIMPs), transforming growth factor β (TGF-β), procollagen type 1 N-terminal propeptide (PINP), galectin-3, osteopontin and soluble interleukin 1 receptor-like 1 (sST2) are the main clinically used indicators of MF [27], which in turn appear to be regulated by epigenetic controllers such as microRNAs (miRNAs). A more sophisticated analysis conducted by a “single-cell”-detector revealed various anatomic districts with different gene networks; therefore, the markers used should be distinguished in relation to myocardial tissue. For example, to identify cardiac valve interstitial fibroblasts, the candidate markers are WNT Inhibitory Factor 1 (WIF1) and the cartilage oligomeric matrix protein (COMP) [28], as well as periostin for epicardium fibrosis [29].

## 3. Cellular and Molecular Pathways Involved in Myocardial Fibrosis

In non-pathological repair mechanisms, the soft balancing between the synthesis and secretion of ECM and the ECM-degrading matrix metalloproteases (MMPs) ensures the equilibrium of the cellular proliferative processes. Upon injury of the cardiac ECM, the activation of several processes inducing the release of pro-inflammatory (e.g., TNF-a tumour necrosis factor-a, IL-1, IL-6, chemokines and reactive oxygen species) and pro-fibrotic factors (e.g., macrophages and lymphocytes), the proliferation of nonmyocytes cells, and scar maturation, allow for the activation of cardiac fibroblasts, by Fibroblast-specific protein 1 (FSP1 and S100A4) converting into myofibroblasts (fibrous tissue), and contributing to the onset of fibrosis by abnormal secretion of collagen 1a1 and other ECM proteins [30]. The persisting wound healing processes are supplanted by fibrotic scar formation that increases the tensile strength of collagen; subsequently, the cells start an adaptive process characterized by activation of genes, in particular α-smooth muscle actin (SMA), a contractile intermediate filament-associated protein, which in turn contributes to cardiac remodelling [30], disturbing cardiac architecture and function through the disruption of conduction of electric signals. In these processes, myofibroblasts within the cardiac scar tissue release proinflammatory and pro-hypertrophic signals, resulting in cardiomyocyte hypertrophy and necrosis followed by replacement fibrosis.

At molecular levels, the ancestor controller of cardiac fibrosis seems to be TGF-β, which is a pro-fibrotic factor in many events linked to the infarct healing process [31]. Indeed, the known transition of fibroblasts into myofibroblasts is mediated by TGF-β [32] and stimulates the synthesis of collagen fibers type I and III, and fibronectin by decreasing collagenase expression and by exacerbating TIMP1 expression.

The dynamic of TGF-β signalling involves TGF-β receptor 2 (TGFβR2) binding, which in turn phosphorylates TGF-β receptor 1 (TGFβR1) and triggers the activation of a plethora of transcription factors, such as SMAD proteins [33] that could finally lead to pro-fibrotic stimuli.

The activation of TGF-β after myocardial infarction is unclarified yet. It seems that many proteases are required for TGF-β activation in the infarcted myocardium, most of them derived from overproduction of reactive oxygen species (ROS) [34]. In reperfused hearts isolated from myocardial ischemia-induced animal, TGF-β exogenous injection seems to attenuate oxidative stress and reduce the release of pro-inflammatory cytokines (tumour necrosis factor (TNF)-α) [35], whereas feline TGF-β injections reduced cardiomyocyte death through p42/p44 mitogen-activated protein kinase (MAPK) signalling [36].

## 4. The Role of MicroRNAs in Myocardial Fibrosis

Growing evidence points to the role of non-coding RNAs and microRNAs (miRNAs) in cardiac fibrosis [37]. This novel class of small non-coding RNA (miRNAs), of 18–26 nucleotides in length, was identified as a key essential regulator of gene expression.

MiRNAs are transcribed from genomic DNA into a long primary transcript, called pri-miRNA, which is larger than a mature miRNA [38], then cleaved by the endonuclease Drosha RNase III, generating an intermediate known as pre-miRNA, which is transported into the cytoplasm by the Exportin-5/Ran-GTP complex [39,40,41] and further processed into double-stranded miRNAs of 22-nucleotide. The endonuclease Dicer forms a mature miRNA [42] that binds to Argonaute proteins within the RNA-induced silencing complex (RISC) and regulates gene expression at the posttranscriptional level by targeting the 3′ untranslated regions (3′-UTR) of mRNA transcripts by Watson-Crick base pairing.

It is now up for debate about the transcriptional modality in which the mature miRNA would bind to their respective 3′UTR. For example, the site in which miRNA is located (exons or introns, or across a splice site) could influence the destiny of pre-mRNA and share with the host gene common regulatory patterns. Indeed, based on specific localization, e.g., a conserved intergenic region, miRNAs can be transcribed and helped by transcriptional machinery of the host gene, whereas a miRNA located in an intron has an independent promoter [43].

MiRNAs involved in cardiac pathology are well-investigated. The main miRNAs governing the pathways for MF activation are the miR-29 family, involved in ECM expression [44], miR-125b governing fibroblast activation, and miR-30 targeting connective tissue growth factor (CTGF), contributing to fibrotic remodelling [45].

Among cardiac microRNAs, miR-21 has been studied in fibrotic processes, being one of the targets of TGF-β signalling; a recent study showed the ability of plasmatic miR-21 levels to predict fibrotic lesions enhanced by levels of collagens I and III, fibronectin expression by targeting RECK, programmed cell death 4 (PDCD4), and transforming growth factor (TGF)-β-signalling factors [46].

Plasmatic miR-21 seems to delineate a specific profile in the regulation of the fibrosis gene program, targeting their downstream mRNA 3′UTR of Jagged1 [47], the phosphatase and tensin homolog deleted from chromosome 10 (PTEN)/AP-1 [48], SMAD family member 7 (Smad7) [49] and sprouty1/2 (SPRY1/2) [50,51].

Since metabolic dysregulation, including glucose metabolism, was recognized in MF progression (through the regulation of TGF-β-mediated hypoxia-inducible factor (HIF)-1α and/or the renin-angiotensin system [52]), it is plausible that miR-21 might play a key role. Recently, miR-21 has been associated with the increase of oxidative stress and has also been associated with glycemic parameters [53,54,55], showing not only a link with the pathogenesis of the onset of cardiovascular complications, but also a growing role as alternative biomarkers coupled to clinical investigations.

MiR-185 has been identified in resident cardiac cells from mice subjected to an experimental model of cardiac fibrosis. In cardiac fibroblasts, the gain of function assay of miR-185 allowed collagen production and profibrotic activation [56]. An in vivo study on mice demonstrated that targeting miR-185 abolished pressure overload induced by cardiac interstitial fibrosis. Mechanistically, it seems that the miR-185-5p binding apelin receptor inhibits their anti-fibrotic effects. It could be speculated that miR-185 expression increases with the degree of fibrosis in virtue of its concomitant increase of pro-fibrotic TGF-β1 and collagen-1 in left ventricular tissue from patients with severe cardiomyopathy [56]. Moreover, it is well known that elevation of miR-185 was associated with selenium deficiency leading to MF; furthermore, pathological changes accompanied by increased miR-185 exerted a reduction of antioxidant properties such as glutathione peroxidase-1(GPx-1) levels in a cellular model of endothelial cells mimicking glycemic variability [57]. A schematic depiction was reported in Figure 2.

## 5. The Advantages and Limitations of Echocardiographic Deformation Imaging in the Assessment of Myocardial Fibrosis

To date, two-dimensional (2D) transthoracic echocardiography (TTE) is the most common cardiac imaging procedure performed in clinical practice, due to its portability, low cost, and patient acceptance. It provides a certain degree of tissue characterisation, especially in the case of thinned and akinetic myocardium and expression of transmural MF [58].

During the last two decades, advances in the echocardiographic imaging have led to the introduction of a 2D speckle tracking echocardiography (STE), an angle-independent technique, which provides diagnostic and prognostic information in several cardiac diseases, such as heart failure, cardiomyopathies and valvular heart disease [59]. The myocardial strain of both ventricles and also the left atrium has been shown to correlate with the degree of MF [60,61].

Echocardiographic strain imaging measures regional and global myocardial function by assessing the deformation (strain) of myocardial fibers in systole and diastole, in longitudinal, circumferential and radial directions and the rate at which this deformation occurs (strain rate) [62,63,64]. Generally, strain (Ɛ) is a dimensionless measure of tissue deformation, expressed as a percentage (%), whereas the strain rate is expressed as unit s^−1^ [64]. The magnitude of strain and strain rate is expressed as a negative value in the case of myocardial fibres shortening, and as a positive value in the case of myocardial fibres lengthening. Consequently, the percentage (%) of strain in shortened contracted fibres is negative, whereas in elongation phase it is positive.

### 5.1. Clinical Utility of Strain Deformation for Tissue Characterization

Unlike CMRI, which defines tissue characteristics through direct observation of the changes in the acquired myocardial tissue images [65], 2D-STE analysis should be considered only indirectly as related to MF, by assessing the impact of the underlying pathology on tissue function [66]. A number of histological and pathophysiological changes affecting the extracellular matrix may impact myocardial mechanics by increasing myocardial stiffness [67,68,69,70]. These changes in cardiomyocyte mechanics are reflected as global or regional impairment in deformation parameters assessed by 2D-STE analysis [71,72,73].

The left ventricular (LV) global longitudinal strain (GLS) is the most commonly used 2D-STE-derived deformation index of cardiac contractility. A number of studies demonstrated a strong inverse correlation between LV-GLS magnitude and the extent of MF in various clinical settings. Notably, LV-GLS has been found to be strongly correlated with the degree of MF in patients with advanced systolic heart failure (HF) requiring heart transplantation [74], in patients with severe aortic stenosis (AS) [75,76,77], in patients with myocarditis [78,79], in patients with hypertrophic cardiomyopathy (HCM) [71], in patients with cardiac amyloidosis (CA) [80,81,82] and finally in patients with dilated cardiomyopathy [72] (see Appendix A). Moreover, during the last two decades, several studies have employed 2D-STE methodology as a diagnostic tool for identifying subclinical myocardial dysfunction and for prognostic risk stratification of various study populations. A significant reduction in LV-GLS magnitude has been associated with worse cardiovascular outcomes in patients with CA [83,84], in patients with HCM [85,86], in patients with AS [87,88], in patients with non–ST-segment elevation myocardial infarction [89], and in pregnant women aged 35 years or older [90]. Qualitative polar maps demonstrating regional strain variations have been demonstrated to be particularly useful for detecting the myocardial areas with the lowest regional strain values corresponding to the greatest myocardial hypertrophy and fibrosis, as in patients with arterial hypertension [91], in patients with HF with preserved left ventricular ejection fraction (LVEF) [92,93], in patients with CA [94], in patients with HCM [95,96], in athletes [97], in patients with AS [98] and finally in patients with nonalcoholic fatty liver disease [99].

Concerning left atrial (LA) strain assessment, 2D-STE deformation imaging provides valuable information about atrial mechanics and its correlation with a range of cardiac conditions [100]. A recent consensus document regarding 2D-STE deformation analysis has been published to standardize the methodology applied for the assessment of the LA chamber [101]. The 2D-STE analysis allows for the detection of LA dysfunction before LA enlargement. The LA reservoir strain, assessed by 2D-STE analysis, is a more direct measure of the intrinsic properties of the myocardium, whereas conventional morphological parameters, such as left atrial volume indexed, just represent an indirect estimation of LA function. LA reservoir strain is a rapid and simple measure that may elucidate the role of atrial function in several pathophysiological conditions, such as mitral valve disease, supraventricular arrhythmias, hypertension, coronary artery disease, HF, atrial stunning and cardiomyopathy. A lower magnitude of LA reservoir strain and increased LA stiffness are correlated with chronicity of LV afterload elevation, causing an increase in LV filling pressures and leading to compensatory cardiomyocyte hypertrophy associated with a significant quota of Int-F [102,103,104]. The interstitial collagen deposition explains the early impairment of LA compliance, even before the LA enlargement. A number of studies have demonstrated the incremental prognostic value of LA reservoir strain over conventional echo Doppler parameters in different clinical settings, such as in patients with AS [105,106,107], in acute ischemic stroke patients without a history of atrial fibrillation [108,109], in pregnant women with new-onset gestational hypertension [110] and in patients with mild-to-moderate idiopathic pulmonary fibrosis [111]. Moreover, atrial cardiomyopathy may exist without atrial fibrillation (AF), can facilitate the development of AF [100,112,113] and is strongly associated with functional impairment of the left atrial appendage [114,115]. Concerning the relationship between LA strain and MF, a significant negative correlation between LA reservoir strain and the degree of LA fibrosis measured on EMB was demonstrated in patients with mitral valve disease undergoing mitral valve replacement or repair [73,74] and in advanced HF patients undergoing heart transplantation [116] (see Appendix A). In addition, a low magnitude of LA reservoir strain was found in patients with high degree of LA wall fibrosis detected on advanced CMRI techniques [117,118].

With regards to the correlation between right ventricular (RV) deformation and underlying tissue characteristics, Lisi M et al. [119] and Tian F et al. [120] demonstrated a direct correlation of RV free wall myocardial deformation with the extent of RV myocardial fibrosis on EMB in patients with severe HF undergoing heart transplantation (Appendix A). From a clinical point of view, the prognostic value of RV-GLS impairment has been demonstrated in the setting of HF with reduced and preserved ejection fraction [121,122] and for predicting chronic HF mortality [123].

### 5.2. Technical Limitation of 2D-STE Analysis

Even if LV-GLS is more sensitive than LVEF, as assessed by conventional 2D-TTE for detecting subclinical myocardial dysfunction, it has not yet been incorporated into everyday clinical practice, due to several limitations which can affect the calculation of strain parameters and their physiological meaning. Indeed, a number of technical factors may influence myocardial strain parameters (see Appendix A). Firstly, 2D-STE analysis is a semiquantitative technique with a learning curve to improve the quality of manual adjustments of the different regions of interest. Indeed, when the region of interest is inappropriately narrow or too wide, it will result in inaccurate strain values [124]. Secondly, it requires good image quality [62] and optimization of frame rates (generally, no less than 40 fps); the tracking quality becomes reduced when the frame rate is too low (for example in the case of tachycardia), due to frame-to-frame decorrelation [125]. Other limitations of 2D-STE methodology are the temporal stability of tracking patterns, due to physiological changes in interrogation angles between moving tissue and the ultrasonic beam during each cardiac cycle [62,126,127], and the intervendor variability, with technical differences among post-processing algorithms [128,129]. Moreover, reductions in myocardial deformation indices are age-related [130], are more commonly detected in males than in females [131], are observed in the setting of the pathological heart rate increase, such as in sepsis [132], are associated with the common cardiovascular risk factors [133,134,135,136,137], and, finally, may be related to the negative inotropic and chronotropic effects of beta-blockers. In addition, it is important to consider that myocardial deformation depends not only on contractile properties of the myocardial fibers (“contractility”), but also on their loading conditions (pre- and after-load), chamber geometry, dyssynchrony, and segment interactions [138,139]. In this regard, a discrete number of studies has been performed to evaluate the influence of chest wall conformation on 2D-STE derived myocardial deformation parameters in various clinical settings [140,141,142]. The chest wall conformation may be assessed by the modified Haller index (MHI), a nonradiological anthropometric index obtained by dividing the latero-lateral (L-L) thoracic diameter (measured by a rigid ruler coupled to a level) by the antero-posterior (A-P) thoracic diameter (measured during conventional 2D-TTE, from the parasternal long axis view, as the distance between the true apex of the sector and the posterior wall of the descending aorta, visualized behind the left atrium) (Figure 3A,B) [143]. Both diameters are measured at the end of inspiration. An increased MHI, due to a narrow A-P chest diameter, was found to be the main anthropometric determinant of the significant reduction in myocardial deformation indices, particularly at the level of basal segments, detected in healthy individuals with a concave-shaped chest wall and/or pectus excavatum (PE) (as noninvasively defined by a MHI > 2.5) [144], in absence of any intrinsic myocardial dysfunction [145,146,147,148,149]. Moreover, individuals with MHI > 2.5 have been found to receive excellent prognoses over mid-to-long term follow-up [150,151,152]. The correlation between MHI and strain parameters is strictly correlated to the degree of the anterior chest wall deformity in subjects with MHI > 2.5, whereas this has not been observed in individuals with normal chest conformation (MHI ≤ 2.5) [145]. Examples of LV-GLS bull’s eye plot patterns obtained in two healthy subjects, with PE and normal chest shape conformation respectively, are depicted in Figure 3C,D.

## 6. The Interplay between Deformation Imaging and Molecular Approaches in the Clinical Decision-Making Process

The heterogenicity of cardiac tissue reflects differences in the expression of extracellular matrix components. Several circulating molecules have been proposed as biomarkers of MF, by virtue of their association with many parameters assessed by EMB, such as myocardial collagen volume fraction (CVF) or myocardial collagen 1 and 3 volume fraction (CVF1 and CVF3, respectively) [153].

In clinical practice, 2D-STE analysis is increasingly being used. This methodology allows for a more detailed assessment of cardiac contractility and may detect a subclinical myocardial dysfunction when LVEF is preserved, in various clinical settings [90,154].

In accordance with the 2016 ESC guidelines [155], the diagnosis of heart failure with preserved LVEF needs several additive serum biomarkers, such as natriuretic peptides, beyond the structural or functional ventricular alterations. In this regard, MF seems to be more frequently observed in HF patients with preserved LVEF than in those with reduced LVEF [156]. Recent studies have focused on the assessment of MF biomarkers in the diagnosis and/or prognostic risk stratification of heart failure (see Section 3). From a pathophysiological point of view, the major limit of these biochemical markers (tissue or circulating) is related to the site of their production which is not always the myocardium, but rather some inflammatory local or systemic processes.

Molecular approaches are in their infancy with respect to serum or plasma biomarkers, in particular miRNAs [37], but they represent the new diagnostic frontier for detection of MF. As miRNAs are circulating molecules, measurable with laboratory methods in blood samples, they actually behave as circulating biomarkers of disease and therefore do not present the limits of the genomic or proteomic approaches. Despite these novelties, the lack of inter- and intra-laboratory standardization make them “putative biomarkers” and a prerogative of research. In addition, miRNAs detection could pave the way for new therapeutic targets specific for MF and comparable to the anti-fibrotic effects of major incretin drugs such as glucagon like peptide-1 (GLP-1) [157,158]

In the great majority of cases, the impairment in myocardial deformation indices has been primarily correlated with pathological myocardial remodeling secondary to MF. Contrary to EMB and cardiac MRI, which directly evaluate the myocardial tissue features in the different pathological diseases, 2D-STE analysis identifies secondary myocardial tissue functional abnormalities without specifying the causes of the underlying histopathological changes. For example, an impaired LV-GLS detected in a patient with significant myocardial hypertrophy is not automatically diagnostic of amyloid infiltration or hypertrophic cardiomyopathy and cannot define the amount of MF eventually present either. However, CMRI and EMB are not easily available for routine use in the clinical practice. Conversely, a 2D-TTE implemented with 2D-STE analysis of all cardiac chambers may also be performed as a first approach at the patient’s bedside and is absolutely non-invasive and repeatable in time [61].

In light of the above-mentioned considerations, we could affirm that abnormally low strain values in a single patient are not necessarily a sign of myocardial dysfunction and/or MF and should always be integrated with other clinical characteristics and diagnostic examinations, due to the possible role exerted by interfering or artifactual factors on 2D-STE results. In this context, molecular approaches might be coupled with 2D-STE analysis, in particular to provide the relationship of the myocardial strain indices with soluble molecules and micro-RNAs. Therefore, the identification of these markers (canonical and non-canonical) could lead to potential strategies for a better comprehension of MF.

From a clinical point of view, a patient with an echocardiographic phenotype suggestive of hypertrophic, infiltrative or dilatative cardiomyopathy with preserved LVEF on 2D-TTE and LV-GLS less negative than -20% on 2D-STE analysis should be treated with cardioprotective and anti-fibrotic drugs, such as angiotensin-converting enzyme (ACE) inhibitors/angiotensin II receptor blockers (ARBs) and/or aldosterone receptor antagonists. On the other hand, the impaired myocardial deformation detected in a healthy individual with anterior chest wall deformity (MHI > 2.5) and normal systolic function assessed by 2D-TTE examination might not be directly considered as a synonym of intrinsic myocardial dysfunction and/or MF, but could be primarily the expression of intraventricular dyssynchrony secondary to the influence of chest wall conformation on the cardiac motion pattern [126].

Due to the high negative predictive value of LV-GLS in different clinical settings [159,160,161,162], a normal LV-GLS value (more negative than −20%) allows clinicians to reasonably exclude a subclinical myocardial dysfunction and possibly even the presence of significant MF. To date, we believe that the incorporation of LV-GLS for clinical decision-making might be of additional value in patients with normal LV-GLS (more negative than −20%) only.

Conversely, due to the lower positive predictive value of a reduced LV-GLS (less negative than −20%) and the possible interference by confounders and/or artifactual factors on 2D-STE analysis, the clinical decision-making process in a single patient with reduced LV-GLS should consider other clinical and conventional echocardiographic variables and a possible indication to other complementary diagnostic techniques [163].

## 7. Conclusions

Considering the complexity of MF, it is critical to identify the relationship between molecular profiles and the functional properties of myocardium assessed by 2D-STE analysis, for a better understanding of the MF pathophysiology. A combined approach evaluating both miRNA-based crosstalk among different cardiac districts, and 2D-STE-derived regional and/or global myocardial strain indices, might improve the prognostic risk stratification of MF patients.

During the critical clinical decision-making process, 2D-STE analysis might not be a direct and univocal expression of intrinsic myocardial dysfunction and should be associated with other clinical and instrumental data. We discussed the innovative role of non-radiological MHI as a “detector” of a particular chest wall conformation. Notoriously, individuals with different patterns of chest shape have different probability of subclinical myocardial dysfunction, such as those with a concave-shape chest wall and/or pectus excavatum. Therefore, misinterpretations might be avoided by the implementation in clinical practice of molecular markers, which may help clinicians to discriminate critical cardiac situations. Further prospective studies should be performed for coupling 2D-STE analysis with molecular or biochemical approaches to validate and strengthen its output data.

## Figures and Tables

**Figure 1 ijms-23-10944-f001:**
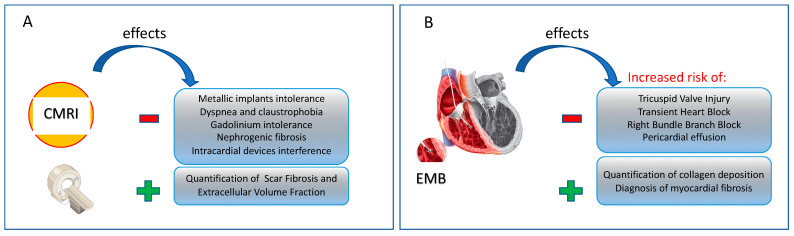
(**A**,**B**) Major negative and positive effects with the use of the non-invasive (**A**) and invasive (**B**) techniques able to identify fibrotic deposition in myocardium. CMRI: cardiac myocardial resonance imaging; EMB: endomyocardial biopsy.

**Figure 2 ijms-23-10944-f002:**
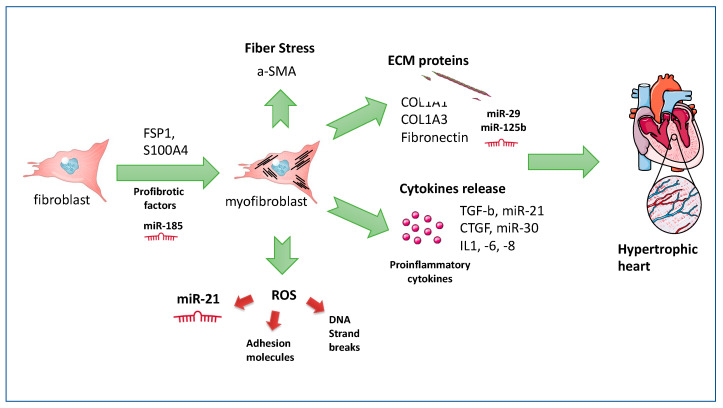
Depiction of the main mechanisms involved in the development of myocardial fibrosis. Upon injury, a fibroblast or other resident cells such as pericytes, endothelial cells, among others, release profibrotic factors including miR-185 to convert into a myofibroblast. The activation of a plethora of damaging agents such as ROS or proinflammatory cytokines and ECM proteins induces myocardial fibrosis and ultimately a hypertrophic heart.

**Figure 3 ijms-23-10944-f003:**
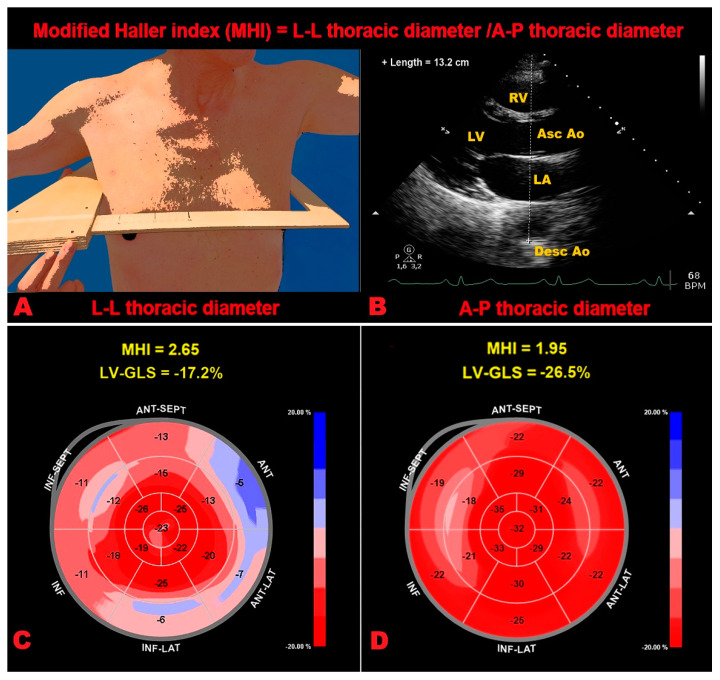
Modified Haller index measurement, obtained by dividing the L-L thoracic diameter (**A**) by the A-P thoracic diameter (**B**). The L-L thoracic diameter is measured with the subject in the standing position and with open arms, by using a rigid ruler in centimetres coupled to a level (the measuring device), placed at the distal third of the sternum, in the point of maximum depression of the sternum. The A-P thoracic diameter is measured, during conventional transthoracic echocardiography, as the distance between the true apex of the sector (the point of entry of ultrasound into the chest) and the posterior wall of the descending thoracic aorta, visualized behind the left atrium. A-P, anteroposterior; Asc ao, ascending aorta; Desc Ao, descending aorta; LA, left atrium; L-L, latero-lateral; LV, left ventricle; RV, right ventricle. Examples of LV-GLS bull’s eye plot patterns obtained in two healthy subjects, with PE (**C**) and with normal chest shape (**D**), respectively. The PE subject (MHI >2.5) was found with a significant impairment in basal myocardial strain (light pink and pale pink segments), moderate impairment in mid myocardial strain (light red segments) and with a normal apical strain (bright red segments); the resultant LV-GLS (−17.4%) was moderately impaired. On the other hand, the subject with normal chest wall conformation (MHI ≤ 2.5) was found to have a uniformly red pattern of the bull’s eye plot, indicating normal regional and global longitudinal deformation of myocardial segments (LV-GLS = −27.2%). GLS, global longitudinal strain; LV, left ventricular; MHI, modified Haller index; PE, pectus excavatum.

## Data Availability

Not applicable.

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
