# Peer review of "Molecular Approaches and Echocardiographic Deformation Imaging in Detecting Myocardial Fibrosis"

_ijms, 2022, doi:10.3390/ijms231810944_

Round 1

Reviewer 1 Report

Dear Authors,

The idea of your  review article is interesting and very topical and  but this study have important  drawback which,  I consider  important  to be corrected

I have added a few specific comments:

1.Although it is a narative review article I think it is important to describe  the aim , the method, the results of the study.

2. State how the data were extracted from included studies. How many studies were included? How many excluded?

3. Summarize, for each subchaper, characteristics of included studies in a table.

4. The recent study about the role of biomarkers in MF : „ Roles of Biomarkers in Miocardial Fibrosis  writting by Ding Y, Wang Y should be mentioned.

5. The phrase „ the principle that inspired our studies.....” line 345  and „our study group” line 351 aren’t correct for this type of article.  It is an review article and the authors talk about their study group.

6. This recent study:”Detection of myocardial fibrosis by speckle-tracking echocardiography: from prediction to clinical applications, M Lisi, M Cameli, GE Mandoli, MC Pastore “ should be , also, mentioned.

7. The conclusion section should be improved.

In conclusion, I recommend accepting the article for publication  with major revision.

Reviewer 2 Report

This topic is interesting for audience who are studied in the assessment of myocardial fibrosis. The author concluded that it is critical to identify the relationship between molecular profiles and the functional properties assessed by 2D-STE analysis, in order to better understand the MF pathophysiology. However, in terms of the title how is the clinical imaging encountered with the molecular evidence…? The clinical imaging is too large, and the structure of this manuscript is not clear and very illogical, which making it very confusing.

Specific comments:

1.In section 2 and 4, a list of references in diagnosis or regulation the occurrence of MF should be summarized into tables.

2.In section 3, schematic diagram should be added to increase readability.

3.In section 5, the title is echocardiographic deformation imaging, the relationship between MF is? The authors focused too much on technicalities on how to assess LA function with echo instead of illustrating the associations of LA strain and myocardial fibrosis. Moreover, the right ventricular deformation correlated with underlying tissue characteristics, which has been reported (10.1016/j.jcmg.2014.12.026, 10.1016/j.jcmg.2021.01.015). Please read carefully. Besides, the technical limitation of 2D-STE needs a bit simplify. Finally, we appreciate the effort that study about the MHI conducted by authors, but the MHI seems not related to the topic of this review.

4.In section 6, some argument needs reference support but there is no reference after that.

Round 2

Reviewer 1 Report

Although it is not a meta-analysis, a review article should present clearly the included studies and the inclusion and exclusion criteria.

Reviewer 2 Report

In this revised manuscript, there are still many details which has nothing to do with the theme myocardial fibrosis, especially in section 5 and 6, and it needed to be revised or deleted. In section 5, the title “the advantage and limitation of echocardiographic deformation imaging in the assessment of MF” may be more appropriate to the topic of this review. In section 6, the title is “the interplay between deformation imaging and molecular approaches”, the interplay is what? and molecular approaches is what? Only miRNAs?

Author Response

As suggested by the Reviewer 2, the old title “Echocardiographic deformation imaging” was replaced by the following new title: “5. The advantages and limitations of echocardiographic deformation imaging in the assessment of myocardial fibrosis”.

Moreover, following sentences were deleted from section 5:

“Reduced levels of reproducibility of STE derived strain measures have been observed with less experienced echocardiographers”.

“The recommendation is that image views, where tracking is insufficient in more than 1 segment, should be excluded from further analysis”.

“A number of considerations would support the importance of this “mechanical theory”. First, the physiological apex-to-base gradient (highest-to-lowest) in left ventricular deformation is usually maintained in PE subjects, both in longitudinal and circumferential directions, thus indicating normal underlying myocardial function. Secondly, the alteration of LV-GLS is generally accompanied by LV global circumferential strain (GCS) impairment in PE subjects, departing from the typical behavior of LV-GLS which usually becomes altered at an earlier stage compared to LV-GCS [152-154]”.

In addition, the title of the paragraph 6 was modified as follows: “6. The interplay between deformation imaging and molecular approaches in the clinical decision-making process”.

Concerning molecular approaches, in the paragraph 6 we added following sentences: “The heterogenicity of cardiac tissue reflects differences in the expression of extracellular matrix components. Several circulating molecules have been proposed as biomarkers of MF, in virtue of their association with many parameters assessed by EMB, such as myocardial collagen volume fraction (CVF) or myocardial collagen 1 and 3 volume fraction (CVF1 and CVF3, respectively) [154]”.

“Recent studies have focused on the assessment of MF biomarkers in the diagnosis and/or prognostic risk stratification of heart failure (see section 3). From a pathophysiological point of view, the major limit of these biochemical markers (tissue or circulating) is related to the site of their production which is not always the myocardium, but rather some inflammatory local or systemic processes”.

“Molecular approaches are in their infancy with respect to serum or plasma biomarkers, in particular miRNAs [37], but they represent the new diagnostic frontier for detection of MF. As miRNAs are circulating molecules, measurable with laboratory methods in blood samples, they actually behave as circulating biomarkers of disease and therefore do not present the limits of the genomic or proteomic approaches. Despite of these novelties, the lack of inter- and intra-laboratory standardization make them “putative biomarkers” and a prerogative of research”.

Round 3

Reviewer 2 Report

I recommend acceptance as the manuscript is well organized and makes the point clearly in this revision.